# Photoinactivation and Photoablation of *Porphyromonas gingivalis*

**DOI:** 10.3390/pathogens12091160

**Published:** 2023-09-14

**Authors:** David M. Harris, John G. Sulewski

**Affiliations:** 1Bio-Medical Consultants, Inc., Canandaigua, NY 14424, USA; 2Department of Periodontics, Rutgers School of Dental Medicine, Newark, NJ 07103, USA; 3Institute for Advanced Dental Technologies, Huntington Woods, MI 48070, USA; 4Millennium Dental Technologies, Inc., Cerritos, CA 90703, USA

**Keywords:** dentistry, laser, periodontal pathogens, photodynamic therapy, photoinactivation, photothermolysis, ultraviolet therapy

## Abstract

Several types of phototherapy target human pathogens and *Porphyromonas gingivitis* (*Pg*) in particular. The various approaches can be organized into five different treatment modes sorted by different power densities, interaction times, effective wavelengths and mechanisms of action. Mode 1: antimicrobial ultraviolet (aUV); mode 2: antimicrobial blue light (aBL); mode 3: antimicrobial selective photothermolysis (aSP); mode 4: antimicrobial vaporization; mode 5: antimicrobial photodynamic therapy (aPDT). This report reviews the literature to identify for each mode (a) the putative molecular mechanism of action; (b) the effective wavelength range and penetration depth; (c) selectivity; (d) in vitro outcomes; and (e) clinical trial/study outcomes as these elements apply to *Porphyromonas gingivalis* (*Pg*). The characteristics of each mode influence how each is translated into the clinic.

## 1. Introduction

Growing concerns about pathogen resistance to chemical antimicrobials have led to a surge of interest in antimicrobial phototherapies that target drug-resistant strains to which the pathogens may not be able to develop resistance. In support of this possibility, Amin et al. (2016) [1] reported no evidence of tolerance development of *Pseudomonas aeruginosa (Pa*) to ten sublethal cycles of antimicrobial blue light and Sabino et al. (2020) [2] tested photodynamic inactivation with methylene blue and red light and found aPDT to be consistently effective against the World Health Organization list of global priority multidrug-resistant pathogens.

Fueled by this possibility, there has developed an enormous literature on numerous approaches to antimicrobial phototherapies using a perplexing array of light sources and wavelengths against scores of different microbial pathogens. In this report, we identify five unique modes for the photoantisepsis of *Pg* and define their characteristics to provide a theoretical framework for understanding this complex topic.

*Pg* is a pathogenic bacterium primarily associated with periodontal disease. The clinical applications directed at photo-eradication of *Porphyromas gingivalis* (*Pg*) will be discussed within the context of dentistry. *Pg* is often chosen as a candidate pathogen for study because it belongs to a class of bacteria known as black pigmented bacteroides that are assumed to selectively absorb light energy. Several different approaches to photoantisepsis can be organized into specific “modes” and we examine *Pg* as a target using these different modes.

### 1.1. Modes of Antimicrobial Phototherapy

At least five unique modes can be differentiated by photon energy and putative chromophores.

Mode 1:Antimicrobial ultraviolet (aUV): 200–300 nm, high-energy photons that target DNA.Mode 2:Antimicrobial blue light (aBL): 400–440 nm, low-power photochemical reaction that targets bacterial and fungal endogenous photosensitizers that generate toxic oxygen radicals.Mode 3:Antimicrobial selective photothermolysis (aSP): 700–1100 nm, high-power, photothermal ablation that targets endogenous chromophores.Mode 4:Antimicrobial vaporization: 1300–11,000 nm, high-power, infrared photons that target intracellular water.Mode 5:Antimicrobial photodynamic therapy (aPDT): 405–980 nm, low-power, photochemical reaction that targets a localized photoactive substance.

### 1.2. Light Can Be Selective

A favorite demonstration of the American physicist and laser pioneer, Dr. Arthur Schawlow (1921–1999), was to inflate a green balloon inside a clear balloon and pop only the green one with a monochromatic laser [3]. A survival strategy of *Pg* is to invade the cytoplasm of host cells [4,5] (Figure 1). With the same “balloon-in-a-balloon” concept, one should be able to selectively destroy intracellular *Pg* leaving the host cell (gingival fibroblasts [6] or gingival epithelial cells (GECs) [7]) intact. Wasson et al. 2012 [8] reduced the intracellular bacterial load of *Chlamydia trachomatis*-infected HeLa cells using 405 nm blue light.

Selectivity is quantified with a concept borrowed from pharmacology, the therapeutic ratio: the light dose that is toxic to the host divided by the light dose that inhibits or destroys the pathology [10]. The larger the ratio, the safer and more effective is the treatment. Several light-based technologies have evolved as potential alternatives or adjuncts to antimicrobial drug therapies. Each technology seeks “selectivity,” a dosage window with an acceptable therapeutic ratio.

There are other advantages to antimicrobial phototherapies. Light provides access to poorly vascularized sites such as ischemic burns and ulcers and “privileged sites” in the oral cavity such as calculus and dentin. Light treatment alone leaves no residuals, is local, not systemic and can eradicate microorganisms instantly or within minutes, while chemical antibiotics can require days to take effect.

### 1.3. Watts, Seconds, Joules and Square Centimeters

In order to describe these modes quantitatively, we need to define specific terms. Power, or the rate that energy is dissipated, is measured in Watts (W). Power density (PD) is easily understood with the following thought experiment. You are a ten-year-old boy with a magnifying glass. The sun is out. You see an unsuspecting ant… A more quantitative example is the beam from a 6 W laser focused into a 1 mm spot. In this case, power density is 6 Watts divided by the area:6 W/π(0.1/2 cm)^2^ = 764 W/cm^2^(1)

Power delivered over time is energy (Watts × seconds = Joules (J)). Power density delivered over time is equal to energy density or light dose (J/cm^2^). Light dose, or surface irradiance, combined with the penetration depth (see below) is analogous to drug dose (mg/kg) as both describe the concentration of a therapeutic agent in a specific tissue volume.

The same light dose can be delivered with a low-power density over a long period of time or with a high-power density in a short period of time. Low-power density irradiation over long time periods drives photochemical reactions. High-power densities result in the rapid accumulation of heat, and in short time periods, can generate photothermal damage [11].

### 1.4. Photon Wavelength and Energy/Biomolecular Target (Chromophore)

Photons are packets of variable amounts of energy that oscillate as they travel through space. High-energy photons vibrate at a higher frequency than low-energy photons. “Wavelength” is the distance a photon travels as it vibrates through one cycle. Since all photons travel at the same speed of light, energy (vibration frequency) determines the wavelength: high-energy ultraviolet photons have shorter wavelengths and lower-energy infrared photons have longer wavelengths. For the purpose of this discussion, we divide the wavelength range of interest, 200–11,000 nm, into seven segments: UVC (100–280 nm), UVB (280–315), UVA (315–400), blue (400–440), VIS (380–700), NIR (700–1300) and H_2_O (1200–11,000). People with normal color vision perceive the color violet in the range 380–450 nm and blue is perceived for 450–495 nm wavelengths. The term “blue light” in this report encompasses “violet-blue.”

### 1.5. Tissue Optics: Absorption and Penetration Depth

When light irradiates tissue, the photons can be reflected, scattered, transmitted or absorbed. No tissue effect occurs without absorption, which leads to photochemical reactions or to significant heating. “Chromophores” are biomolecular components that absorb light energy. Each chromophore has a unique absorption spectrum, a description of an entity’s ability to absorb a photon as a function of the photon’s wavelength. Absorption results in the conversion of photon energy into radiant, rotational or vibrational energy. Chromophores that transfer energy to produce toxic reactive oxygen species (ROS) are termed “photosensitizers.”

Different chromophores absorb strongly in different wavelength regions. For example, water absorption dominates the region >1200 nm and these photons are absorbed at the tissue surface with minimal penetration depth. Also, an optical window exists in tissue in the near-infrared (NIR) between about 700 and 1200 nm [12,13] (Figure 2) where there is less absorption and photons can penetrate more deeply into tissues. This provides the ability to treat large tissue volumes (the therapeutic window). The depth a specific wavelength can travel into tissue is quantified as the penetration depth, which is equal to the depth at which the energy density is attenuated to 1/e (37%) of the surface irradiance (Table 1).

## 2. MODE 1: Ultraviolet Inactivation through DNA Damage

A comprehensive account of the origins and evolution of ultraviolet (UV) and violet-blue light phototherapies is provided by Enwemeka et al. (2021) [15] and Dai et al. (2012) [16] review of antimicrobial UVC, specifically.

### 2.1. Mechanism of Action

In the late 19th century, Nobel laureate Dr. Niels Ryberg Finsen developed a method for using concentrated ultraviolet (UV) light to treat lupus vulgaris [17], which is caused by *Mycobacterium tuberculosis,* known now for its intrinsic drug resistance and antibiotic tolerance. Although the phototherapy became common in clinical practice, it was several decades before the mechanism of action was well understood. It is now accepted that UVC radiation is mostly absorbed by nucleic acids in RNA and DNA [18,19] which can generate toxic photochemical reactions. An immediate effect of DNA-damaging UV is a transient inhibition of DNA synthesis. Intrinsic repair mechanisms may partially reverse DNA damage, known as microbial reactivation [20]. Cytotoxic DNA lesions affect metabolism and disrupt the ability to reproduce [21]. Peak et al. (1984) [19] obtained action spectra for lethality in *Escherichia coli* (*Ec*) that closely matched the average DNA spectrum of Setlow (1974) [22]. Figure 3 presents more recent data showing the normalized disinfection rates for *Ec, Pa and Staphylococcus epidermidis* (*Se*) and a portion of the DNA absorbance spectrum (derived from Matsumoto et al. 2022 [23]). Ultraviolet irradiation in the 200–300 nm wavelength range is germicidal. This range includes both UVC and UVB wavelengths. There is a peak in the absorption spectrum of DNA at 254–260 nm that also corresponds to the greatest bactericidal efficacy [24,25]. Antimicrobial UV wavelengths off-peak require a higher light dose to achieve the same log reduction.

### 2.2. UVC Dosimetry

*Pg* has about the same sensitivity to UVC as other oral pathogens, although Henry et al. (2008) [26] reported that a *uvrB*-defective mutant strain of *Pg* was more sensitive to UVC than the wild-type W83 *Pg* strain. Near the peak sensitivity, Metzger et al. (2007) [27] tested endodontic pathogens, *Pg, Enterococcus faecalis (Ef), Streptococcus sanguinis (Ss), Fusobacterium nucleatum (Fn)* and *Lactobacillus brevis* (*Lb*) to 254 nm light. *Pg* was 99.9% eliminated at 2–6 mJ/cm^2^. Takada et al. (2017) [25] observed with 265 nm light the complete eradication of *Pg*, *Fn*, *Ss* and *Streptococcus mutans* (*Sm*) at 17 mJ/cm^2^. The power density was 1.7 mW/cm^2^. An international standard (DIN EN 14897:2007-09, 2007) for germicidal efficacy for industrial devices is 40 mJ/cm^2^ for UVC.

### 2.3. Penetration Depth

The penetration depth of UVC light in Caucasian skin is only about 2 μm at 250 nm (Table 1). Metzger et al. (2007) [27] measured a light dose of 2–7 mJ/cm^2^ for 100% surface elimination of *Pg* but also observed “shielding” in biofilms. Only four cells in the light path were sufficient to block UVC light, requiring a 10× greater light dose to effectively treat a multilayer biofilm. They concluded that a germicidal UVC dose required in vivo may be orders of magnitude higher than that in vitro.

### 2.4. Selectivity

A phototherapy can be selective if there is an acceptable therapeutic ratio. UVC treatment advocates the claim that there is selectivity since mammalian cells have repair mechanisms that limit UV toxicity, although bacteria, including *Pg,* also have DNA repair mechanisms [28,29]. Dai et al. (2012) [16] assert that bacteria are more sensitive to UVC than mammalian cells. They acknowledge that UVC treatment of a tissue volume is not feasible due to the shallow penetration depth. This means that a therapeutic dose in deep tissues (more than a few cell widths) may be toxic at the surface. It is also possible that UV germicidal wavelengths of 222 nm or 310 nm are less toxic to normal cells but have less germicidal efficacy. It is uncertain if the therapeutic ratio is better.

From above, it can be assumed that the in vitro germicidal dosimetry for *Pg* inactivation is in the range of 2–17 mJ/cm^2^ for UVC irradiation. Dai et al. [16] conclude that a germicidal UVC dose required in vivo may be orders of magnitude higher than that in vitro, so assume that a minimally effective clinical light dose is about 20 mJ/cm^2^. The guideline for UVC exposure by the American Conference of Governmental Industrial Hygienists suggest an exposure limit value of 3 mJ/cm^2^ [30]. The therapeutic ratio calculated from these data is 3/20 or 0.15. This therapeutic ratio applies to a specific set of estimated conditions and should not be generalized to all conditions.

## 3. MODE 2: Blue Light Inactivation through ROS Generation (aBL) [15,31,32,33,34]

Photo-eradication of pathogens with blue light has generated considerable attention because of the potential germicidal efficacy and safety. The current status is thoroughly reviewed by Wang et al. (2017) [34] followed by Enwemeka et al. (2021) [15] and Leanse et al. (2022) [32]. Leanse et al. predict that blue light photoinactivation may become the “Magic Bullet” to thwart microbe antibiotic resistance.

### 3.1. Mechanism of Action

The current thinking is that blue light is absorbed by endogenous photosensitizers, molecular components of a bacterium that absorb photons and transfer the photon energy to generate toxic reactive oxygen species (ROS) [33,35,36,37,38,39,40,41]. The absorption of photon energy raises the photosensitizer from the ground energy state to an excited state. From the excited state, the energy is transferred through either a Type I or Type II reaction to produce free ROS that induce membrane damage and intracellular damage wherever the photosensitizers are located. Detailed reviews of the photochemical reaction pathways are provided by Hamblin and Hasan (2004) [42], Enwemeka et al. (2020) [15], Rapacka-Zdończyk et al. (2021) [43] and Leanse et al. (2022) [32].

The location of photosensitizers is important for effective photoinactivation. Results from Kato et al. (2018) [39] indicate that photosensitizers located in the outer layer of the cytoplasmic membrane inactivate membrane function and the strength of photoinactivation is dependent on their affinity to the cell membrane. This implies that membrane-bound photosensitizers [36] are a primary target for aBL inactivation. Rapacka-Zdończyk et al. (2021) [43] describe both gram-positive and gram-negative bacterial membrane structures that are likely involved in photodynamic inactivation. They also identify intracellular elements that may likewise be involved in photoinactivation. The reactions leading to inactivation may be different for different species. Chui et al. (2012) [44] suggest that for *Pg*, blue light may not be directly bactericidal but rather inhibits growth by suppressing the expression of genes concomitant with chromosomal DNA replication and cell division. In this case, ROS production results in DNA damage [45].

Whole-cell absorbance measured with diffuse reflection spectroscopy (Harris et al. 2016, Appendix) [9] represents the summed absorbance of all endogenous chromophores. The high concentration of porphyrins within certain pathogens is reflected in their whole-cell absorbance spectrum (Figure 4 and Figure 5). In Figure 4, the porphyrin heme structure “signature” of a Soret absorption band around 410 nm and Q-band peaks at longer wavelengths is apparent [46]. The blue-light whole-cell spectrum of *Staphylococcus aureus* (*Sa*) from 400 to 430 nm approximates the action spectrum for photoinactivation (red circles) [47], both of which reflect the Soret-band absorption by endogenous photosensitizers.

*Pg* whole-cell absorption also demonstrates a Soret peak at 410 nm with an absorption coefficient of about 1000 cm^−1^ (Figure 5) [9]. Colony growth inhibition of *Pg* at 405–425 nm is in the range of 0.3–62 J/cm^2^ [32,36,48,49,50,51]. Higher power densities require less time to achieve the same light dose. Fukui et al. (2008) [48] found higher PD more efficient against *Pg* in the range of 30–100 mW/cm^2^.

Many authors conclude that blue light inactivation does not occur at wavelengths beyond about 430 nm for most pathogens studied [47,48,49]. *Pg* has a peak of 410 nm like other pathogens tested but also shows strong absorption up to about 650 nm (Figure 5). This accounts for *Pg’s* greater sensitivity to ROS stimulation at wavelengths >450 nm. Kim et al. (2013) [49] reported a 525 nm light-dose-suppressed growth of *Pg* biofilm. With 400–500 nm broadband exposure, the photoinactivation dose for *Pg* was 16–62 J/cm^2^, whereas inactivation for *Sm* and *Streptococcus faecalis* (*Sf*) was 3× to 10× higher at 159–212 J/cm^2^ [50]. The response of *Pg* to the bactericidal effect of argon laser (488 nm and 514 nm) also indicates that it has greater absorption in that portion of the spectrum. In the study by Henry et al. [52], a light dose of 35 to 80 J/cm^2^ from an argon laser inhibited growth of *Pg* biofilm. Data from Izzo and Walsh (2004) [53] at 455 nm and 625 nm indicate that the bactericidal mode at these longer wavelengths may be thermal.

### 3.2. Penetration Depth

The review by Leanse et al. (2022) [32] estimates the tissue penetration depth of blue light to be about 400 μm. However, this may be an overestimate since the penetration depth in the skin is only 90 μm at 400 nm and 150 μm at 450 nm (Table 1). Song et al. (2013) [54] treated *Pg* at 400–520 nm with a halogen curing lamp. They estimated a 30–45 μm depth of kill and much lower efficiency in biofilm vs. planktonic state due to quorum interactions and variance in oxygen distribution in the biofilm.

### 3.3. Selectivity

Apparently, aBL is far less toxic to mammalian host cells when compared to microbial pathogens. Leanse et al. (2022) [32] examined published data from 15 various studies including both in vitro and clinical studies. A toxic dose is estimated as the median value of 110 J/cm^2^ from their summary of host cell damage_._ If the in vivo photoinactivation of *Pg* requires a 10× increase in light dose in vitro (2–7 J/cm^2^) [27], then in vivo inactivation would be in the range of 20–70 J/cm^2^. The therapeutic ratio is calculated to be 110:20–70 or in the range of 1.6–5.5. This selectivity is a result of the high concentration of photosensitizes within certain bacteria relative to mammalian cells. Bacteria are rich in cytochromes, porphyrins and other photosensitizers [32,55].

## 4. MODE 3: Thermal Ablation through Antimicrobial Selective Photothermolysis (aSP)

As defined by Anderson and Parrish (1983) [56], selective photothermolysis “relies on selective absorption of a brief radiation pulse to generate and confine heat at certain pigmented targets. An absolute requirement is that the targets have greater optical absorption at some wavelength than their surrounding tissues.”

Selective photothermolysis (SP) has revolutionized the field of dermatology to produce a “cosmetic laser industry” for selective removal of vascular anomalies such as telangiectasias, including acne rosacea and spider angioma [57], tattoo pigment removal [58] and permanent hair removal by selective destruction of hair follicles [59]. This concept can also be applied for the selective destruction of periodontal pathogens.

### 4.1. Mechanism of Action

*Pg* scavenges heme from the environment that first attaches to the outer membrane. Once transported across the membrane, it localizes within the cytoplasm where it supports several metabolic functions [60]. *Pg* synthesizes endogenous porphyrins known as tetrapyrroles including iron protoporphyrin IX that aggregate on the cell surface [61]. These chromophores in *Pg* account for its high absorption in the VIS and NIR. Since the materials available in the environment vary, the visual colors of *Pg* colonies can vary. The whole-cell absorption spectrum of *Pg* from 390 nm to 1100 nm (Figure 5) shows the combined absorption of all the chromophores that are intracellular or attached to the outer membrane [9]. All chromophores contribute to light absorption but, unlike aBL, chromophores do not need to be photosensitizers.

The mode of selective photothermolysis requires increased power densities and short interaction times to achieve thermal damage (Figure 6). Pulsed dental lasers like a 6 W Nd:YAG or Er:YAG emit the 6 Watts in very short-duration high-peak power pulses repeated at a certain repetition rate. For example, typical parameters used for laser sulcular debridement for the pulsed Nd:YAG laser are 100 μs duration pulses with a peak power of 1000 W/pulse (100 mJ per pulse) repeated at 20 Hz equals 2 W average power. As these pulses exit a 320-micron diameter optical fiber, the peak power density during the pulse is 1.2 × 10^6^ W/cm^2^. This value of power density causes a rapid increase in temperature and the 100 μs interaction time confines thermal damage to the target. Above 60° to 70 °C, structural proteins such as collagens are denatured and above 70° to 80 °C, nucleic acids are denatured. Soft tissue heated to 70–100 °C will undergo protein denaturation, leading to "coagulation necrosis." Above 100 °C, there ensues the vaporization of water with rapid expansion that separates or ablates tissues. These temperatures must be maintained for a certain period of time in order to cause thermal damage. The exposure time and temperature necessary to accomplish thermal damage in a *Pg* colony is based on the thermal capacity of the colony, modeled as: molar entropy (ΔS) = 585.1 J/mol K and the molar enthalpy (ΔH) = 2.9 × 10^5^ J/mol [9,62]. With this model, a 111 °C exposure temperature for a 100-μs exposure time (pulsed Nd:YAG, Er:YAG, or CO_2_ laser) or an 84 °C temperature for 100-ms (pulsed diode laser) will cause thermal damage (Figure 6).

### 4.2. Selectivity

The absorption coefficient of a substance, μ_a_, describes the relative rate of attenuation of light as it travels through that substance, in this case, a bacterium or its medium. Larger values of μ_a_ represent a higher probability of absorption, and better absorption implies greater efficacy. A therapeutic ratio for photoablation of *Pg* in the periodontal ligament can be estimated from the difference in absorption coefficients of *Pg* and its host environment at a specific wavelength; larger differences mean greater selectivity. Figure 7A shows the relative absorption spectra (μ_a_ as a function of wavelength) of three periodontal pathogens, *Pg, Prevotella intermedia* (*Pi*) and *Prevotella nigrescens* (*Pn*), and spectra for two of the host environments, the periodontal ligament (PL) and water. The periodontal ligament is modeled as normal healthy connective tissue at 75% water, blood volume 1.7% (35 μM HbO_2_ and 60 μM Hb), 15% collagen and 5% extracellular ground substance and cellular components [63]. At 810 nm, the ratio of absorption coefficients is 10:0.03 or 333 and at 1064 nm, the ratio is 7.7:0.18 or 43. There is also a significant difference (>100) in absorption coefficients of *Pg* compared to dentin (see Section 5.1, Figure 8).

The optimum wavelength entails the greatest difference in optical absorption between the target and the host. For excellent specificity, the ratio of target-to-tissue coefficients (therapeutic ratio) should be on the order of 10 or greater, but SP may be achievable with ratios as low as 2 [56]. The optimum wavelength is also selected to provide the maximum depth where SP can destroy a target. It can be seen in Figure 7A that there is a difference of up to 1000× between *Pg* and PL in the range of 500–900 nm. Due to increased water absorption in the range of 900–1000 nm, the ratio remains 10× for *Pg* but is less selective for the other two microbes. The window opens again at 1000 nm to about 1100 nm, where Table 1 indicates penetration depths of more than 2 mm. The window closes for longer wavelengths beyond 1300 nm that are strongly absorbed by water. These windows are illustrated in Figure 2 for *Pg* and *Pn*. Clinically, this determines the wavelength with the safest therapeutic ratio and a maximum depth of kill.

Selectivity depends on environmental conditions [64]. Henry et al. (1995) [35] observed that *Pg* phototoxicity to an argon laser (488 and 514 nm) was influenced by the hemin concentration in the culture, which also influences porphyrin content [64]. This was studied in detail by Harris et al. (2016) [9]. The absorption spectra of *Pg* cultured in increasing concentrations of hemin (0.5–15 μg hemin/mL) were measured with diffuse reflection spectroscopy (Figure 7B). There was a strong correlation between the hemin concentration and the level of visual pigmentation in the colonies (illustrated by the graduated bar in the inset). The hemin strongly asserted its effect on the absorption coefficients in the visible range from 400 nm to about 900 nm but had a minimal effect on the longer wavelengths in the near-infrared range from 900 to 1200 nm, which includes the 980 nm diode and the 1064 nm Nd:YAG lasers. Heme availability influences the absorption coefficient, hence the therapeutic ratios computed for *Pg* above will depend on those environmental conditions at 810 nm but less so at 1064 nm.

### 4.3. Depth of Kill

Given these optical and thermal properties of the pathogens and the host tissue, Harris and Reinisch (2016) [63] constructed a mathematical model for the surgical scenario of “laser sulcular debridement” and ran a simulation to illustrate the depth of kill in a virtual periodontal ligament. In the simulations, 50 μm diameter colonies were placed at 1, 2, 3 and 4 mm depths. An optical fiber was inserted in the periodontal pocket, light pulses were delivered and the fiber was passed around the tooth four times at 1 cm/s. A video of sequential instantaneous thermal profiles illustrates accumulating selective thermal damage in the colonies. One of the videos shows a comparison of the thermal profiles for three dental lasers. *Pg* colonies have absorption coefficients equal to 10 cm^−1^ at 810 nm, 7.7 cm^−1^ at 1064 nm and 13,000 cm^−1^ at 2940 nm, which is the same as water. Both the 810 nm diode laser and the 1064 nm Nd:YAG laser achieved a 2–3 mm deep kill zone for *Pg* and *Pi* in the periodontal ligament without surface damage. Accumulated background heat from multiple passes increased the depth of the kill zone. The 2940 nm Er:YAG laser ablated the surface of the periodontal ligament but did not influence the 1 mm-deep *Pg* colony. Another video illustrates photoablation of three different pathogens with different absorption coefficients. In that comparison, the absorption coefficient of the target determined the depth of kill.

## 5. MODE 4: Cell Lysis through Explosive Vaporization

### 5.1. Mechanism of Action

Water absorption dominates over all other soft tissue absorbers at wavelengths longer than about 1250 nm (Figure 8). The erbium lasers and carbon dioxide lasers are near peaks in the water absorption spectrum. Because *Pg* is mostly water, one can consider water to be the intracellular chromophore at those wavelengths. Absorption of photon energy by a water molecule raises the molecule to a higher energy vibration mode, effecting a transition from liquid to vapor and the rapid expansion lyses the cell [65]. Akiyama et al. (2011) [66] examined with electron microscopy inoculated root surfaces and *Pg* cultures after Er:YAG laser exposure and concluded that ablation was by photothermal vaporization. The Sethasathien group (2022) [67] conducted an in vitro investigation of the bactericidal effect of the Er,Cr:YSGG laser on *Aggregatibacter actinomycetemcomitans* (*Aa*) and *Pg*. They also indicated that the mechanism may be thermal evaporation or a photothermal reaction. Similar results of effective bacterial reduction are reported following CO_2_ irradiation of *Pg* suspensions [68] and titanium discs [69].

**Figure 8 pathogens-12-01160-f008:**
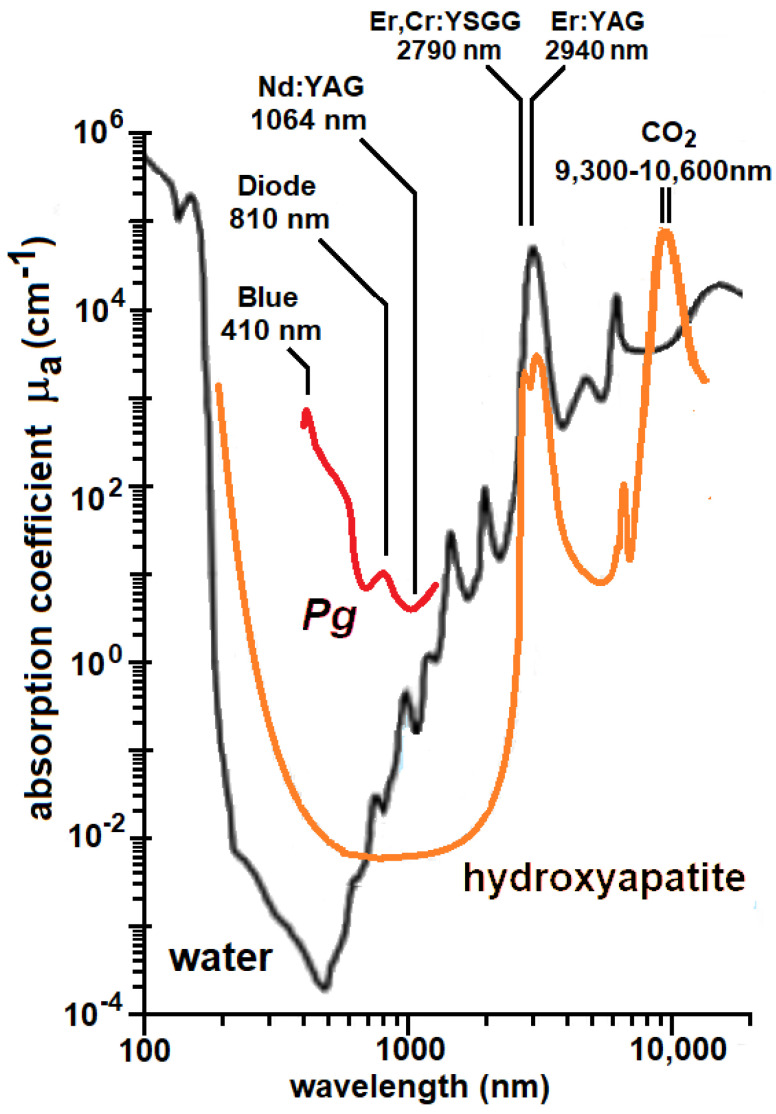
Water, carbonated hydroxyapatite and *Pg* absorption spectra, and the wavelengths of different light sources. Hydroxyapatite curve adapted from Parker et al. (2007) [70].

Dentin is composed of 47% inorganic carbonated hydroxyapatite, 33% organic material and 20% water [71]. Figure 8 shows the absorption peaks of hydroxyapatite at about 2900 nm and 9600 nm, which are close to the output wavelengths of 2790 nm Er,Cr:YSGG and 2940 nm Er:YAG and the 9300 nm CO_2_ lasers. Because of the high affinity of hydroxyapatite at these wavelengths, these lasers have been used to ablate hard tissues.

### 5.2. Selectivity and Depth of Penetration

The soft tissue penetration depth is only 2 μm at 3000 nm and 20 μm at 10,000 nm (Table 1) and the absorption depth in dentin is in the range of 6.7–12 μm from 2790 nm to 10,600 nm [72]. *Pg* is known to invade dentin either from the root surface in the periodontal pocket or internally from the root canal. The effect of mode 4 irradiation on *Pg* is no different than the effect on any other living cell that contains water and is surrounded by an aqueous environment. High-power density erbium and CO_2_ lasers will ablate the surface layer of soft tissue or dentin and its contents indiscriminately and have minimal effect on deeper tissues. We may assume, then, that the therapeutic ratio is close to zero. Consequently, mode 4 demonstrates no selectivity in periodontal tissues or dentin and the depth of kill is restricted to the surface [63].

## 6. MODE 5: Cell Lysis through Drug/Light Interaction (aPDT)

### 6.1. Mechanism of Action

The mechanism is similar to that described for aBL, except the photosensitizer is delivered exogenously and is then selectively accumulated in the target tissues. When irradiated with light of the proper wavelength, the energy absorbed by the photosensitizer is transferred through a Type I or Type II photochemical reaction to generate ROS that can oxidize biomolecules and destroy cells. Research on the photochemical pathways leading from photon absorption to inactivation is detailed in Hamblin (2016) [73] and Rapacka-Zdończyk et al. (2021) [43].

Photodynamic therapy (PDT) was originally devised as a nonsurgical treatment for certain types of cancer [74]. Figure 9 illustrates the procedure. A photosensitizer, in this case, hematoporphyrin derivative (HpD), is delivered intravenously and after 24 h has localized in the cancerous cells of a squamous carcinoma just below the jaw as evidenced by HpD’s fluorescence (Figure 9A). HpD has a Q-band absorption peak at 620 nm, so light at 620 nm from an argon-pumped dye laser is delivered via an optical fiber inserted into the center of the tumor (Figure 9B). Light plus HpD generates ROS and, at 48 h posttreatment, necrosis is localized to the tumor mass (Figure 9C). Specific criteria for PDT include the characteristics of the photosensitizer and matching light source:Selective uptake or binding of exogenous photosensitizer by cells of the target tissuesSelective wavelength for photoinactivation○Maximize absorption by photosensitizer○Optical window to maximize depth of kill 650–1000 nmPresence of oxygen for ROS production

Early researchers suggested that the basic procedure could be modified for the photo-eradication of localized infections, antimicrobial PDT (aPDT; also known as antimicrobial photodynamic inactivation, aPDI, and photodynamic antimicrobial chemotherapy, PACT) [75,76,77]. Various combinations of photosensitizers, light sources and protocols have since been studied in dentistry as an antimicrobial treatment for periodontitis [78,79,80,81,82], peri-implantitis [83,84,85,86], and root canal disinfection [87,88,89]. There is yet to evolve a standard protocol [90]. However, there are essential differences between PDT and aPDT. PDT uses intravenous injection and systemic delivery of a photosensitizer, whereas aPDT relies on topical application and local diffusion. The photosensitizer in PDT is designed to localize in tumors, whereas the photosensitizer in aPDT needs to localize in pathogens. Hamblin (2016) [73] provides a set of criteria for photosensitizers to be used against bacteria:Nontoxic in the darkGood quantum yields of ROSHigh molar absorption coefficient in red and near-infrared spectrumSelective for microbial over mammalian cellsCationic charges

Investigations of photodynamic therapy and *Pg* in particular have utilized lasers at 405, 532, 630–690, 805–830 and 905–980 nm and LEDs ranging from 390 to 480 and 565 to 671 nm. Among the photosensitizers examined in research involving *Pg*, the most extensively studied to date have been indocyanine green [91,92,93], methylene blue [94,95,96] and toluidine blue [97,98,99]. Other photosensitizers have included: aminolevulinic acid [100], azulene [101], β-NaYF_4_:Yb^3+^, Tm^3+^@TiO_2_ (termed UCNPs@TiO_2_) [102], BLC 1010 and BLC 1014 [103], cadmium telluride nanocrystals [104], captopril-protected gold clusters [105], cationic amino acid-porphyrin conjugate 4i [106], Chlorin-e6 [103,107,108], chloro-aluminum phthalocyanine [109,110], Coumarin 6 [111], Curcumin [112,113], curcumin-NP [114], cyanidin-3-glucoside [115], doxycycline [116], erythrosine [115,117], graphene silver polymethyl methacrylate [118], lysine-porphyrin conjugate 4i [119], meta-tetra(hydroxyphenyl)chlorin (mTHPC) [120], oxyhemoglobin-based oxyHb@IR820 [121], phenothiazine chloride [122,123,124], phloxine B [125], Photodithazine [126], phycocyanin [127], protoporphyrin IX [35], purpurin-based photosensitizer [128], Radachlorin^®^ [129], riboflavin [130], rose bengal [131,132], ruthenium-based photosensitizers [133], safranine O [134], sinoporphyrin sodium [135], temoporfin [136], tolonium chloride [137] and zinc phthalocyanine [138].

Certain photosensitizers have exhibited some degree of antimicrobial effects on a variety of micro-organisms relevant to dentistry in the absence of photoexcitation, including methylene blue (MB), toluidine blue-O (TBO), indocyanine green, rose bengal and erythrosine. In general, though, the antimicrobial effects of photosensitizers tend to be more pronounced when the agent has been exposed to low levels of irradiation [139].

The Bhatti group [140,141] studied the destruction of *Pg* using TBO and 632.8 nm red light from a helium-neon (HeNe) laser. The TBO localized in the outer membrane and, to a lesser extent, the inner cytoplasmic membrane. Inactivation was the result of lipid peroxidation and/or protein–protein cross-linking and a resultant decrease in membrane fluidity. Electron microscopy showed membrane condensation and vacuolization of cells.

### 6.2. Selectivity and Depth of Penetration

Topical application of a photosensitizer is probably as effective as topical application of a chemical antibiotic, although the great advantage of aPDT is to avoid resistance. Repeated applications of aPDT utilizing methylene blue and 670-nm illumination to methicillin-resistant *Staphylococcus aureus* (MRSA) could not promote microbial resistance [142]. Rapacka-Zdończyk [43] present evidence of “tolerance” (changes in susceptibility) to aPDT and aBL following a series of sublethal cycles but did not consider it “resistance” since photoinactivation was still possible with an increased light dose.

The light sources selected for aPDT are mostly from 635 to 810 nm. The photosensitizer determines the best wavelength, which may be outside the therapeutic window. Photosensitizers commonly tested for aPDT include methylene blue with an absorption peak of 668 nm [143], toluidine blue O with an absorption peak of 630 nm [144] and indocyanine green (ICG) with broad absorption from 640 nm to 940 nm [145]. The penetration depth of 600–700 nm in the skin is 0.5 to 0.75 mm (Table 1). Wavelengths in the 1000–1200 nm range may penetrate tissues more deeply but are not absorbed efficiently by these photosensitizers. The depth of kill is limited to a few hundred microns with delivery of the photosensitizer through simple diffusion. Rogers et al. (2018) [128] found aPDT “fully efficacious” at removing in vitro *Pg* biofilms to an average depth of 23 μm. However, data indicate that although aPDT is effective in vitro on zirconia implants [146], titanium blocks [147] and in planktonic suspensions, it cannot disrupt root canal biofilms [148,149].

The purpose of aPDT is to sensitize the target for photodynamic destruction. *Pg* is naturally sensitized and is easily destroyed with the other modes described above. In fact, its sensitivity can be enhanced simply by increasing availability to heme in the environment (Figure 7B). However, there is overwhelming evidence that *Pg* can be further sensitized with an array of photosensitizers including MB, TBO or ICG and responds to aPDT as well as, if not better than, other pathogens [94,97,150,151].

This review did not capture any reports of adverse effects of aPDT from clinical or animal [152,153,154] studies. Solarte et al. (2022) [145] studied the bactericidal efficacy of ICG plus a broad-spectrum light source. They also tested for cytotoxicity of aPDT on human gingival keratinocytes. Values that killed the bacteria (60 J/cm^2^) also caused morphological damage to normal cells. The level of toxicity was related to both the ICG concentration and irradiation time. Without threshold values, their data cannot provide an accurate therapeutic ratio and results cannot be generalized to other aPDT photosensitizers and light sources.

There are three primary sources of potential toxicity of aPDT. The light source, especially a diode laser, can accumulate thermal damage if left immobile or delivered with a high-power density in an attempt to shorten treatment time [155]. The photosensitizer can in itself be toxic [145]. The combined photosensitizer plus light is designed to be synergistic and toxic dosimetry is yet to be defined. This is an area that warrants further study.

## 7. Summary and Translation to Clinical Practice

### 7.1. Mode 1: Antimicrobial Ultraviolet (aUV):

Mechanism of action: high-energy photons that target DNAOptimal wavelengths: 200–300 nm with peak efficacy near 265 nm, outside therapeutic windowPenetration depth: depth of penetration 2 μmSelectivity: a therapeutic ratio of less than one (3/20 or 0.15) is not selective

Clinical studies and translation to practice: UVC is successfully employed for an operating room environment and air disinfection [15] and is proposed for surgical sites [156] and wound disinfection [16]. Implant and surgical instrument sterilization are applications of ex vivo aUV sterilization in dentistry that do not require selectivity. However, Han et al. (2018) [157] compared the decontamination of zirconia discs using UVC versus dry heat. Heat was more efficient and there was higher adhesion of *Pg* to the zirconia surface after UVC sterilization than after dry heat. Metzger et al. (2007) [27] suggested that an appropriate dental application in endodontics would be to include aUV irradiation in the protocol for disinfection of root canals. In vitro studies have demonstrated some efficacy [24,27].

The risk/benefit of this treatment mode appears to be different from that of other phototherapies. With other modes, a toxic dose results in local tissue damage; however, with the aUV mode, a toxic dose could result in a potentially lethal malignancy since damage to nucleotides encompasses human DNA. This represents a significant disadvantage of using aUV light for disinfection in vivo [38].

### 7.2. Mode 2: Antimicrobial Blue Light (aBL)

Mechanism of action: violet-blue visible light that targets bacterial and fungal endogenous photosensitizers that generate ROSOptimal wavelengths: 390–440 nm with peak efficacy at 405–410 nm, wavelengths outside therapeutic windowPenetration depth: depth of penetration 90–150 μmSelectivity: a therapeutic ratio in the range of 1.6–5.5 is estimated from in vitro studies

Clinical studies and translation to practice: Blue light therapy has been practiced in dermatology for more than 100 years [158]. The fact that visible light is quite harmless for human cells represents a benefit with regard to medical applications. Potential fields of operation include the disinfection of air and surfaces [159]. Suggested dental applications include general disinfection of the oral and nasal cavities [15], root canal disinfection [160] and dental implant disinfection [161]. Soukos et al. (2015) [162] irradiated dental plaque in vivo at 455 nm, 70 J/cm^2^ and received reductions in *Pg* and *Pi.*

This mode of photoantisepsis is currently experimental with few, if any, published clinical studies in dentistry. Specific applications in dentistry need to be developed and safety and efficacy validated in clinical trials. Zhang et al. (2023) [161] have developed an interesting application of aBL for treating peri-implantitis, so far tested only in implants placed in the tibia of rabbits. A 410 nm LED is incorporated into a zirconia implant and provides low-level, continuous blue light irradiation. A similar application in endodontics might be a light-transmitting root canal fill. Enwemeka et al. suggest that aBL dosimetry values can be reduced significantly if the light is pulsed to synchronize with the aBL photochemistry. Pulsing is timed to “pump” ROS production [15].

### 7.3. Mode 3: Antimicrobial Selective Photothermolysis (aSP)

Mechanism of action: high-power density, thermal ablation that targets endogenous chromophoresOptimal wavelengths: for *Pg* 700–1200 nm, within therapeutic windowPenetration depth: depth of *Pg* kill is 2 mm at 810 nm and 1064 nm depending on environmental conditionsSelectivity: for *Pg*, the therapeutic ratio can be higher than 100 but >10 from 400–1200 nm in both soft tissues and dentin under most environmental conditions

Clinical studies of aSP applied to black pigmented, red complex periodontal pathogens: Dental lasers were introduced as surgical instruments and early users suggested that they could also be used for “reduction of bacterial level” [163,164,165]. Published results from in vivo sampling from periodontal pockets following surgical debridement protocol with the 1064 nm Nd:YAG laser [166,167,168,169,170,171,172] and diode lasers at 685, 805, 808, 810 and 980 nm [173,174,175,176,177,178] universally show immediate reduction of *Pg* often maintained for 3–6 month follow-ups. Several dental laser systems have also been evaluated for their ability to disinfect root canals. Saydjari et al. (2016) [179] reviewed 22 endodontic studies of bacterial reduction with 1064, 810 and 980 nm lasers. They conclude that all are effective at bacterial reduction and that pigmented bacteria were more sensitive to photodestruction.

High-powered dental lasers operating in mode 3 have been cleared for antibacterial marketing claims. From 10 September 1999 through 31 March 2023, the U.S. Food and Drug Administration (FDA) has cleared for marketing certain Class 4 diode dental lasers (7 at 810 nm and 7 at 980 nm) for “reduction of bacterial level,” without specifying the bacterial species involved. One Class 4 Nd:YAG dental laser received clearance on 12 July 2019 for “reducing bacteria on the dentin surface,” based on an in vitro study involving *Bacillus subtilis, Escherichia coli* and *Bacillus stearothermophilus*. No other laser types have received such clearances.

Translation to practice: Unlike the extensive basic research on aBL, the clinical applications came first and translation had already transpired. This is because high-powered dental lasers have a long history as surgical instruments used to achieve surgical outcomes to which bacterial reduction is subordinate. In an actual procedure such as sulcular debridement and removal of the pocket epithelium or root canal shaping, the antimicrobial action is incidental to those surgical objectives. Determining the dosimetry range for effective bacterial reduction from those clinical data is challenging. No single standardized protocol exists and all relevant parameters are rarely published. Nevertheless, there is sufficient evidence that the bacterial load of *Pg* and other oral pathogens is significantly reduced during these procedures using the Nd:YAG laser and many of the dental diode lasers with output wavelengths within the therapeutic window.

The clinician needs to understand this mode of bacterial reduction to better optimize it during a procedure. For example, for soft tissue surgical procedures, diode laser users are often instructed to “initiate the tip” of the optical fiber by coating it with a layer of carbon. The result is absorption of the light energy as it exits the fiber creating a “hot tip” that turns out to be an excellent tool for dissecting soft tissues. However, in this scenario, heat is transferred to the tissue and bactericidal light is blocked.

This mode has proven to be effective at bacterial reduction and demonstrates the greatest depth of kill within the therapeutic window. Although bacterial reduction is a welcomed side effect, it is subordinate to a surgical procedure. Protocols designed specifically for aSP need to be developed. Another barrier to translation is that high-powered dental lasers are costly and require significant training.

### 7.4. Mode 4: Antimicrobial Vaporization

Mechanism of action: high-power density, thermal ablation that targets intracellular waterOptimal wavelengths: >1300 nm, outside the therapeutic windowPenetration depth: depth of penetration 5–20 μm in both soft tissues and dentinSelectivity: for *Pg* in dentin or soft tissues—none.

Reviews of clinical trials and studies: A review by Sgolastra et al. (2012) [180] of five randomized, controlled clinical trials concluded that there was no evidence of the effectiveness of Er:YAG lasers for the treatment of periodontal disease. The review by Zhao et al. (2014) [181] of 12 controlled clinical trials also found no evidence that erbium laser protocols were better than scaling and root planing (SRP) alone for improvement in clinical signs of periodontitis. The Świder et al. (2019) [182] review found Er:YAG laser application shows no significant effect on bacteria in the periodontal pockets in the long term. However, some individual clinical studies do show significant bacterial reduction [183]. Erbium lasers are efficient surgical tools used for sulcular debridement in the treatment of periodontitis and peri-implantitis. Several investigators have also established in vitro their efficacy for disinfecting implant surfaces [184,185]. CO_2_ lasers are also efficient at implant disinfection without causing surface alterations [69,186].

Cheng et al. (2012) [87] determined with electron microscopy and cultures that an Er:YAG laser can completely remove *Pg* biofilm from ex vivo root canals and up to 200 μm into lateral dentinal tubules.

Translation to clinic: Jurič and Anić. (2014) [88] reviewed the surgical challenges associated with using lasers for cleaning and disinfecting the root canal system and evaluated outcomes following Er:YAG and Er-Cr:YSGG laser treatments. They concluded that the erbium lasers are efficient at the removal of the smear layer and both wavelengths have bactericidal effects.

Erbium lasers are currently in use for hard-tissue procedures, root canal management and periodontal pocket debridement, although several literature reviews [180,181,182] suggest that these lasers may not be the most effective choice for periodontitis and peri-implantitis treatment. Linden and Vitruk (2015) [187] provide a detailed protocol for using the 10,600 nm CO_2_ laser to treat peri-implantitis. Miyazaki et al. (2003) [168] compared the results of Nd:YAG and CO_2_ laser treatment of periodontitis in 18 patients. They reported a significant decrease in *Pg* and an improvement in clinical signs with Nd:YAG but not with CO_2_. However, erbium lasers are apparently the optimal system for removing biofilms and the use of erbium lasers is indicated for canal shaping and disinfection. Due to the shallow depth of penetration, residual infection often remains in accessory (side) channels of the root canal that could be accessed with a dual-wavelength system having shorter wavelengths within the therapeutic window.

### 7.5. Mode 5: Antimicrobial Photodynamic Therapy (aPDT)

Mechanism of action: photochemical reaction that targets a localized photoactive substance to produce ROSOptimal wavelengths: the optimum wavelength depends on the absorption characteristics of the selected photosensitizer, preferably within the therapeutic windowPenetration depth: the depth of kill is limited by wavelength selection and diffusion of the photosensitizerSelectivity is indeterminate because of the lack of data on toxic dose vs. therapeutic dose. The lack of reports of toxicity of aPDT may indicate high selectivity

Reviews of clinical trials and studies: Dai et al. (2009) [188] provided one of the first reviews of aPDT. When the Dai group wrote their review in 2009, they noted that aPDT applied to treat dental and oral localized infections was a rapidly growing clinical application. At that time, most of the data were from in vitro studies and some companies were beginning to run trials of aPDT systems for dental applications. They conclude that the available systems have some efficacy but are not yet optimized. Meisel and Kocher (2005) [189] saw promise in these early studies and encouraged further development and testing of aPDT applied to periodontal and endodontic disease.

Since 2009, there have been hundreds of clinical studies published just on the dental applications of aPDT. The Meimandi et al. (2017) [190] review analyzed 16 studies of treatment for periodontitis comparing SRP vs. SRP + aPDT. Half of the studies showed that aPDT had an incremental effect on improving clinical signs and bacterial reduction. Peron et al. (2019) [90] reviewed 4 of 49 studies of aPDT for periodontitis. aPDT was effective at reducing *Pg*, *Aa, Fn* and *Pn* as well as causing decreased inflammation. All four studies used a 660–662 nm diode but four different photosensitizers. The authors noted substantial heterogeneity in the aPDT parameters. Another review of aPDT for periodontitis by Vohra et al. (2016) [191] of seven controlled and randomized trials came to an identical conclusion: aPDT is effective as an adjunct to SRP but there is considerable heterogeneity of aPDT parameters among studies. Fraga et al. (2018) [192] reviewed aPDT studies applied to peri-implantitis. Only three studies met their strict inclusion criteria, also with a mixture of aPDT parameters: photosensitizers were TBO (activated with 810 nm and 690 nm diode lasers) and phenothiazine chloride (activated with a 660 nm diode laser). They concluded that aPDT is effective at bacterial load reduction (*Pg, Aa,* and *Pi*) in peri-implantitis. A review by Muhammad et al. (2015) [193] searched “endodontic disinfection with laser” and generated 306 articles from 1982 to 2014. Although some studies show promising results, they considered the approach to be still controversial. They provided a useful discussion on finding the proper place for aPDT within a medically sound protocol for canal debridement, shaping and disinfection (including accessory channels) prior to obturation. The most recent review by Gholami et al. (2022) [194] analyzed 60 randomized, controlled trials of aPDT in periodontal (42), peri-implant (6) and root canal infections (12). They concluded that aPDT is effective as an adjunctive treatment to reduce the bacterial load but could not recommend specific irradiation parameters “due to heterogeneity among studies.”

Translation to practice: Amidst this assortment of aPDT parameters and applications have emerged a few commercial products for aPDT-based systems with protocols and delivery systems to treat periodontitis, peri-implantitis and endodontic infections. The photosensitizers are mostly based on MB in solutions or gels that are injected into the periodontal or peri-implant pocket or into the root canal. Activation is with fiberoptic-delivered light from a 660–670 nm diode laser. These commercial systems are marketed outside the United States with claims of efficacy based on a substantial literature showing positive outcomes.

Several authors have indicated potential drawbacks to the use of aPDT. It is unknown how aPDT affects benign oral flora. Treatment may lead to phototoxic or photoallergic unwanted side effects and dominance of a single resistant species [189]. There is concern about the potential cosmetic and metabolic side effects of residual photosensitizers in the pocket and surrounding tissues [101,155]. To mitigate this concern, aPDT protocols should include techniques to localize diffusion and for routine removal of the photosensitizer after treatment.

As a note of caution, aPDT is still considered to be experimental in the United States. As of 31 March 2023, no aPDT device has received U.S. FDA marketing clearance. The plethora of protocols needs to be culled and safety better defined with studies that are designed to identify levels of toxicity.

## 8. Conclusions

*Pg* is an appropriate candidate to examine photoantisepsis in general since all modalities identified in this report are effective at reducing its numbers or inhibiting its growth. Targeting the reduction of *Pg* will remove its immune subversion strategies that benefit cohabiting species [195]. In the case of selective thermal ablation, *Pg* may act as a heat sink and transfer thermal damage to the rest of a mixed-species colony. With its various evasion strategies, *Pg* has been resistant to the immune system and chemical antibiotics. Colonies located in dental plaque, calculus, dentinal tubules and the cytoplasm of host cells are all sites that are inaccessible by neutrophils or amoxicillin but accessible by light.

Light-based antisepsis is currently a multifaceted and very active area of research in engineering, microbiology and clinical laboratories. Several research groups around the world are testing a wide variety of blue light protocols, photosensitizers and dental laser delivery systems. Some researchers are testing combined modalities in an attempt to increase efficacy. For example, Amaroli et al. (2020) [196] tested in vitro combined aPDT and aSP with an 810 nm diode laser. Chui et al. (2013) [125] tested in vitro aBL combined with aPDT and noted a greater log reduction from the combined treatment compared to aBL alone.

The five modes of *Pg* inactivation or ablation are localized and minimally invasive light-based modalities that show promise in reducing and controlling the bacterial load in the oral cavity. Understanding the mechanisms, characteristics and limitations of each modality will allow the clinician to apply the proper technique to achieve the desired clinical outcome. This report suggests a framework for the organization of a complex literature. This review provides an overview for the graduate student, experienced researcher or clinician to fill in any gaps in their knowledge base and to stimulate interdisciplinary research.

## Figures and Tables

**Figure 1 pathogens-12-01160-f001:**
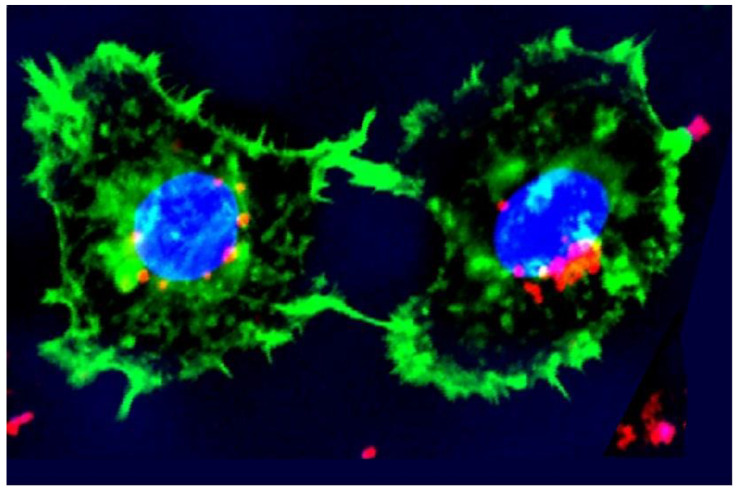
Immunohistofluorescence imaged with confocal microscopy of invasion of gingival epithelial cells (GECs, green) by *P. gingivalis* (*Pg*, red). Once the *Pg* are intracellular, they cluster at the nuclear membrane (blue). Using light, it may be possible to selectively remove the *Pg* and leave the GECs intact. Invasion assay provided by Stephan R. Coats, PhD, University of Washington. Reproduced with permission from Harris, Jacques and Darveau [9] 2016, Wiley.

**Figure 2 pathogens-12-01160-f002:**
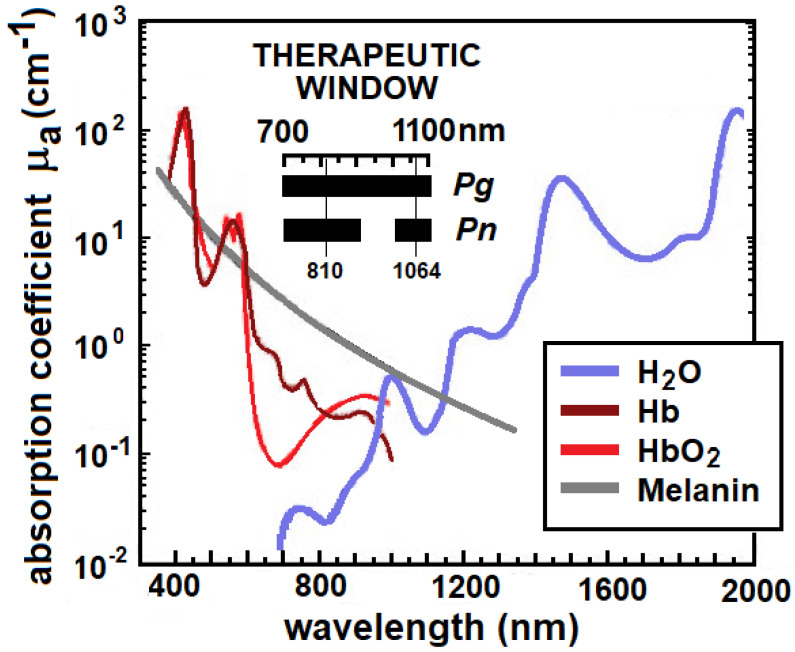
The therapeutic window is a band of wavelengths where tissue chromophores (H_2_O, Hb, HbO_2_ and melanin) have minimal absorption and a therapeutic light dose has maximum depth of kill. The horizontal bars show the 700–1100 nm window for *P. gingivali*s and a narrower window for *Prevotella nigrescens* (*Pn*), which has lower absorption coefficients than *Pg* (see Section 4.2, Figure 7A).

**Figure 3 pathogens-12-01160-f003:**
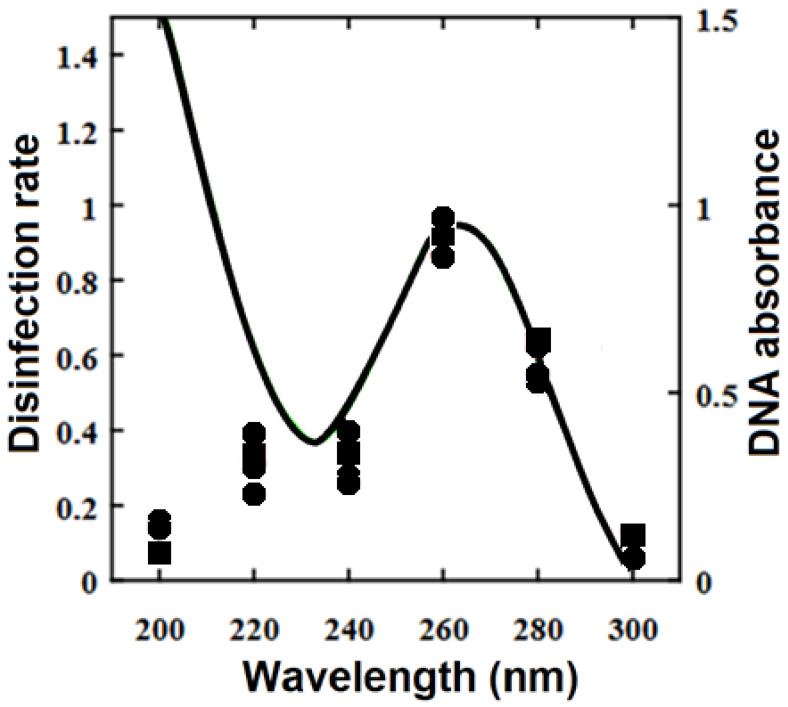
Comparison of normalized disinfection rate of *E. coli, P. aeruginosa and S. epidermidis* (data points) with the absorption spectrum of DNA (curve). Redrawn with permission from Matsumoto et al. (2022) [23].

**Figure 4 pathogens-12-01160-f004:**
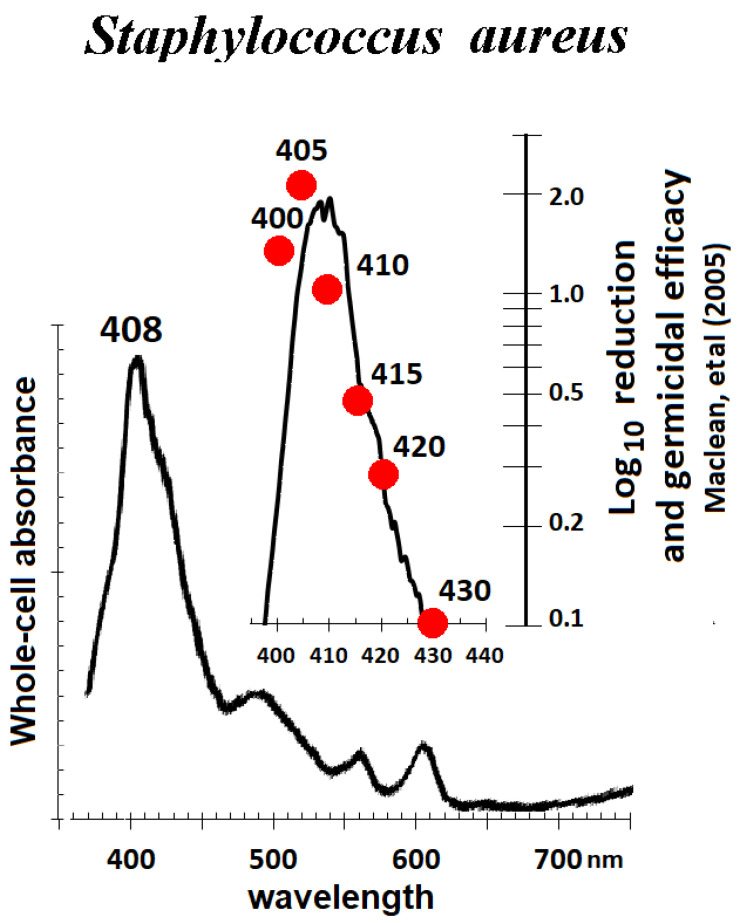
Comparison of diffuse reflection whole-cell spectra of *S. aureus* with bactericidal data. There is a high correlation between the blue light “Soret band” absorbance and germicidal efficacy. Adapted from Harris (2023) [46].

**Figure 5 pathogens-12-01160-f005:**
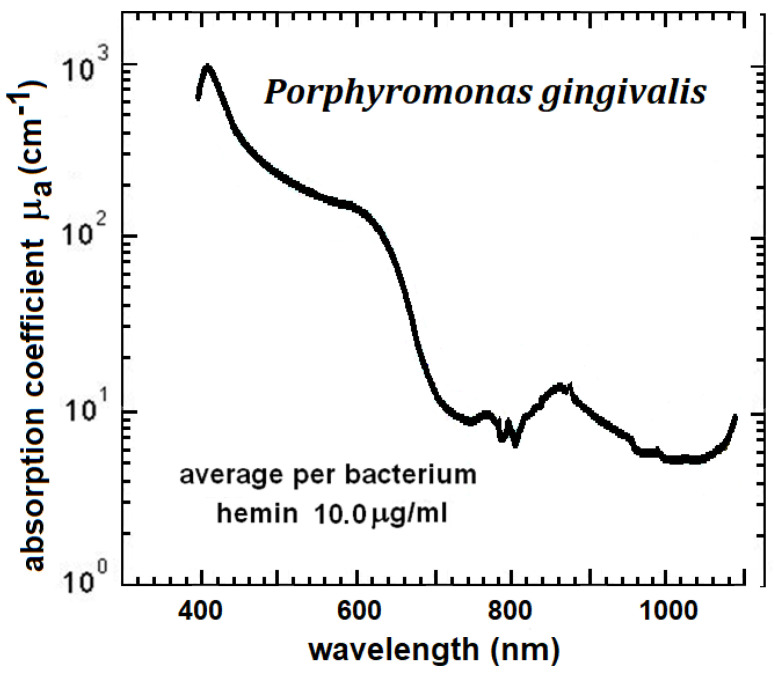
Absorption spectrum of *Pg* obtained with diffuse reflection spectroscopy. *Pg* demonstrates high absorption in the VIS and NIR due to an abundance of endogenous chromophores. Data from Harris et al. (2016) [9].

**Figure 6 pathogens-12-01160-f006:**
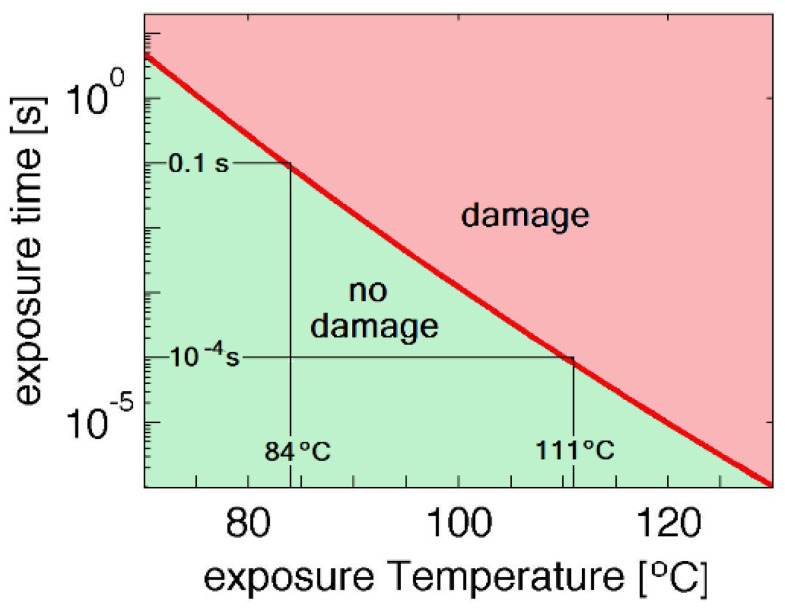
Minimum temperature maintained for an exposure duration required to cause thermal damage in a *Pg* colony. Reproduced with permission from Harris, Jacques and Darveau, [9], 2016, Wiley.

**Figure 7 pathogens-12-01160-f007:**
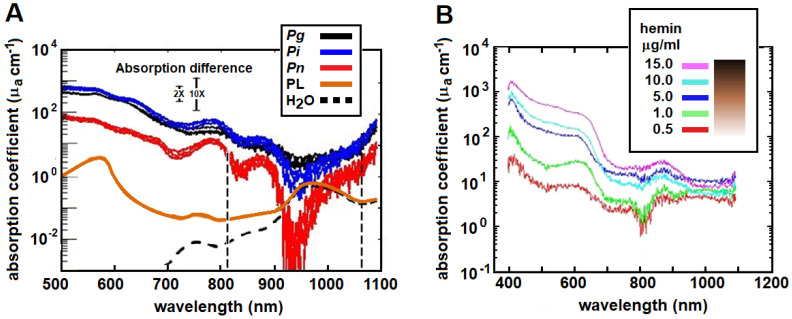
(**A**) Absorption coefficients for *P. gingivalis (Pg), P. intermedia (Pi) and P. nigrescens (Pn)* compared to absorption of the periodontal ligament (PL) and water. Vertical dashed lines are positioned to illustrate the differences at 810 nm (diode laser) and 1064 nm (Nd:YAG laser). (**B**) Absorption spectrum for *P. gingivalis* cultured in a gradient of hemin concentrations. Hemin concentration influenced *Pg* absorption from 390 nm to about 950 nm but not at longer wavelengths. The inset illustrates the gradation in visual pigmentation associated with each concentration. Data adapted from Harris et al. (2016) [9], periodontal ligament from Harris and Reinisch (2016) [63].

**Figure 9 pathogens-12-01160-f009:**
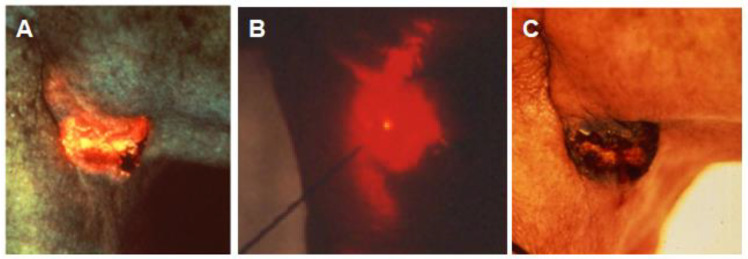
Photodynamic therapy (PDT) for the treatment of solid tumors. (**A**). Localization of the photosensitizer, HpD, within the tumor is indicated by its fluorescence. (**B**). Light at 620 nm from an argon-pumped dye laser is delivered to the center of the tumor by a fiber optic. (**C**). The tumor is selectively destroyed. Harris, D.M; Hill, J.H. “Photoradiation Therapy.” Case presented at the 40th Annual Midwest Conference, Chicago Medical Society, Chicago, IL, March 1984.

**Table 1 pathogens-12-01160-t001:** Penetration depth (1/e of incident energy density) for light of different wavelengths (WL) into fair Caucasian skin from Anderson and Parrish (1981) [12]. Values at 3000 nm and 10,000 nm are soft tissue penetration depths from Sakamoto et al. (2018) [14].

WL (nm)	250	280	300	350	400	450	500	600	700	800	1000	1200	3000	10,000
Depth (μm)	2	1.5	6	60	90	150	230	550	750	1200	1600	2200	2	20

## Data Availability

No new data were created.

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
