# Peer review of "Photoinactivation and Photoablation of Porphyromonas gingivalis"

_pathogens, 2023, doi:10.3390/pathogens12091160_

Round 1

Reviewer 1 Report

Dear editor

This is a review of in vivo and in vitro studies about the Photoinactivation and Photoablation of Porphyromonas Gingivalis in dentistry. To the best of my knowledge, this is the first study that detail the use of Laser and photo sources as Photoinactivation and Photoablation of Porphyromonas Gingivalis in dentistry. I found that the paper was well-written, it has good flowability, and it may improve our knowledge about this topic. 

However, I was wondering why the authors did not perform this review as systematic review following the adequate guidelines (PRISMA) to elaborate such study and obtain PROSPERO registration.

One more concern, the introduction of the paper is too poor, and seems likely to be out of the general context of the paper. I suggest to the authors to elaborate a new introduction of their well performed review.

I really appreciate the time and effort that the authors have put into this paper.

Minor editing of English language required.

Author Response

REVIEWER #1

This is a review of in vivo and in vitro studies about the Photoinactivation and Photoablation of Porphyromonas Gingivalis in dentistry. To the best of my knowledge, this is the first study that detail the use of Laser and photo sources as Photoinactivation and Photoablation of Porphyromonas Gingivalis in dentistry. I found that the paper was well-written, it has good flowability, and it may improve our knowledge about this topic. 

Response: Reviewers are tasked with a critique of a manuscript, yet seldom inform authors what they like. Thank you for your positive comments.

However, I was wondering why the authors did not perform this review as systematic review following the adequate guidelines (PRISMA) to elaborate such study and obtain PROSPERO registration.

Response: Systematic reviews answer a narrow question through detailed and comprehensive literature searches. This is a narrative review that provides the authors' subjective overview of current approaches to photoantisepsis.

One more concern, the introduction of the paper is too poor, and seems likely to be out of the general context of the paper. I suggest to the authors to elaborate a new introduction of their well performed review.

Response: LINES 31-42

We agree that the Introduction needed improvement. It has been slightly reorganized and we added a more focused statement of the objectiv

Author Response

REVIEWER #2

Comments The review entitled “Photoinactivation and Photoablation of Porphyromonas Gingivalis“ by David M Harris and John G Sulewski lists different light-associated methods for the inactivation of Porphyromonas gingivalis. While I think the topic is highly relevant for the field, I suggest a major revision of the manuscript:

Response: Thank you for your time and efforts to improve this review.

  1. 2: I would suggest writing the species epithet not with a capital letter in the title.

Response: Done

  1. 13: Is ultraviolet mutagenesis a valid term? Shouldn’t it rather be ultraviolet light/radiation mutagenesis?

Response LINE 45. We are uncertain about this comment. Light is radiation. We use the term here as a label for Mode 1. To avoid confusion this label is changed to “antimicrobial ultraviolet (aUV)” and Mode 3 selective photothermolysis (SP) is changed to aSP to provide a more consistent nomenclature. 

  1. 19: Please remove the word count in braces at the end of the abstract.

Done

  1. 28: the words “methylene blue” are linked to sciencedirect.com from Elsevier. Is that desired? Also, the font differs here from the rest of the text.

Done

  1. 29: Is the capital letter for the word “Global” correct here?

Changed to lower case.

  1. 32: “light treatment leaves no residues”: the authors might be correct concerning the application of light alone. I highly doubt that aPDT for example with methylene blue does not lead to residues at least for the short term. However, the authors should bear in mind that there are also photosensitizers that in fact do not lead to staining residues.

Response. This is a valid point and we have modified the statement LINE 81. We acknowledge the problem of residual photosensitizer on LINE 729-730, “There is concern about cosmetic and metabolic side-effects of residual photosensitizer’s in the pocket and surrounding tissues[150,206].”

  1. 36-37: The authors use here a different font. By the way, the font of the references in brackets differs also from the rest of the text throughout the manuscript.

Done

  1. 47: Where is Fig.1 derived from? Is it own work of the authors or from another publication? Furthermore, more experimental details should be given to understand the figure from a scientific point of view.

Response: LINE 66-71: We have modified the caption to provide more detail on the methodology and provided information on the origin of the image.

  1. 58: Please include light/radiation after Ultraviolet-C. Please do that throughout the manuscript.

Response: Please see comment re. line 13 above.

  1. 71-72: It seems as there is a part of the thought experiment missing.

Response LINE 88: An ellipsis (. . .) has been added to indicate the missing conclusion.

  1. 106: Again, the question: is the figure own work? Where are the data derived from? This is not clear to the reader.

Response: The plot of tissue chromophores is originally from Figure 2, Boulnois (1986) and has been reproduced enough to become generic. The “Therapeutic window” inset illustrates discussion from section 4.2.

  1. 128: there is a grey shade behind the name “Niels Ryberg Finsen”. Why is that?

Repaired

144: the figure is not adequately described and in general of bad quality. The reader cannot differentiate; which organism is which in the data points. Furthermore, the citation is included in the picture – why is it not cited according to the journal guideline?

Response LINE 165: The image quality and content is improved and the source is properly identified. The data points overlap and that is an important message from the figure.

  1. 173-174: Again, the font differs here from the rest of the manuscript.

Done

  1. 197-203: Please provide literature for the statements.

Response LINES 294-303: This section has been revised and relocated in Section 4.1

  1. 206: The authors mention the lifetime of singlet oxygen in the context of aPDT. However, different photosensitizers have might have different modes of action (keywords type I and type II reaction) that do not (only) lead to the generation of singlet oxygen but also to the generation of other ROS. Can the authors include this to their manuscript as well as information (and references) on the lifetime of ROS other than singlet oxygen?

Response LINES 222-229: This section has been revised. The reader is referred to recent reviews that provide an in depth discussion of the biochemical reactions leading to ROS generation and photoinactivation.  There are several places like this in the manuscript where we could digress into more detail but avoid the tendency to provide an overview.

  1. 244: The references are in round brackets – please use a consistent citation throughout the manuscript as this is the case several times (like in ll. 335, 414, 417, 580)

Done

  1. 295-296: “mol” not “mole” and “s” not “sec”. Please use SI units.

Done

  1. 299-308: Again, it is not clear where the data of fig. 6 and 7 are derived from.

Response ALL FIGURES: Sources have been included in the captions.

  1. 395-407: This section rather describes the therapeutic procedure and not the mode of action. This should be changed.

Response ENTIRE SECTION 6.1: This edit greatly improves the discussion, thank you. This section is revised according to the comment

  1. 408-412: Again, the font differs.

Done

  1. 408: It should rather be “uptake or binding” as some photosensitizers might not be taken up by the cells – or at least there is no literature evidence that really proves that photosensitizer uptake is necessary.

Response LINE 446: We agree! In fact, Rapacka-Zdoncyzk (2021) suggests that the PS may only need to be in the vicinity of the target.

  1. 422: The figure caption is not in style with the other figure captions.

Caption changed to conform

  1. 426-438: References are missing.

Response LINES 476-492: This section has been revised and 50 additional references added.

  1. 433: The authors list hydrogen peroxide as photosensitizer. However, I do not think that this is a typical photosensitizer.

Response: There were two studies that evaluated hydrogen peroxide as PS, Mahdi et al (2015) and Odor et al (2020).

  1. 443: What do the authors want to say with “subablative” in this context?

Response LINE 497: “subablative” changed to “subinactivation.”

  1. 518: The question mark seems odd to me in this context.

“?” Removed

  1. 637-651: It seems the authors put more focus on the negative aspects of aPDT in contrast to the other methods mentioned. The manuscript would profit from a more balanced review in the last section concerning all topics.

Response: This is another observation that improves the manuscript, thank you. LINES 560-564 already discuss problems with UV, LINES 580-582 discuss issues with aBL, LINES 632-636 are issues with aSP and LINES 671-674 with Mode 4.

  1. 652-671: The conclusions are not focused enough on the review itself. Please elaborate more on gaps in the research that might need to be filled in the future. With that in mind: what should future researchers focus on?

Response: The next comment helped improve the conclusions  - LINE 746-753. The negative aspects mentioned above were incorporated into revisions of Section 7 as “barriers to translation.”

Why does the review matter to the reader?

These potential phototherapies are a multifaceted and active area of research in engineering, microbiology and clinical laboratories. This report suggests a framework for the organization of a complex literature. We expect this review to be an overview for the graduate student, experienced researcher or clinician to fill in any gaps in their knowledge base and to stimulate interdisciplinary research.

Reviewer 3 Report

Dear Colleagues,

The Review by David M. Harris and John G. Sulewski summarises knowledge about phototherapy modalities. 5 modes of phototherapy are discussed including UVC, blue light, photothermolysis,vapourization and aPDT, in the light of the probability of inactivation of oral pathogens, especially Porphiromonas gingivalis. All these modes were characterized with light wavelength, mechanism of action, the depth of light penetration, selectivity and clinical trials.

The Authors provide a lot of useful information, like absorption spectra of bacterial cells, tissues and important substances as natural chromophores , dentin and so on. A lot of data on photoinactivation of oral pathogens and side effects and restrictions of these modes are also considered.

Nevertheless, there are some shortcomings, and the first is about the presentation of the mechanism of UV killing.

Line 123.   Ultraviolet inactivation through mutagenesis.  

Line 127    mechanism of action: ultraviolet mutagenesis

Line 132   it is now accepted    and references of 1980 and 1984 years.

Of course, UV is a powerful mutagen, and some mutations are lethal, but DNA damages like thymine dimers , DNA cross linking and others, are not mutations. If not repaired, they induce replication and transcription arrest, leading to the death of bacterion. To convert DNA damage to mutations repair processes are necessary, some of them being error prone, conducting trans- lesion synthesis . I recommend to refer5 to a fundamental book by E.C. Friedberg, G.C. Walker and W. Siede. DNA repair and mutagenesis. 1995. ASM Press, Washington, DC. Or more recent publications.

Line 183   Mode 2: blue light inactivation through ROS generation (aBl)

       The title of the section is not a good fit. This section discusses not blue light, but rather the whole cell absorption spectra, and the effect of various wavelengths on bacteria employing their own chromophores as photosensitizers for ROS production.

Line 313     Two host tissues,the periodontal ligament and water     What are two tissues?

Line 317     The therapeutic ratio can estimated… The phrase is not correct, and 19 mg/ml = of what?

All this block is not understandable, though the Authors describe their own experiments.

Line 437     Many photosensitizers tested on oral pathogenes are listed in this block without any references, and among them there are H2O2 which is not PS.

Line 447      advantage of aPDT is to avoid resistance    . This is a controversial statement, because some groups have demonstrated the possibility of significant and inherited changes in susceptibility to aPDT and blue light after series of sublethal cycles. (See a review Rapacka-Zdo ´nczyk A, Wo ´zniak A, Michalska K, Piera ´nski M, Ogonowska P, Grinholc M and Nakonieczna J (2021) Factors Determining the Susceptibility of Bacteria to Antibacterial Photodynamic Inactivation. Front. Med. 8:642609.

doi: 10.3389/fmed.2021.6426)

Line 470     They also tested for cytotoxicity of human gingival keratinocytes.  Inaccurate phrase.

Line 537 and beneath.     The lasers operating in Mode 4 are discussed in the section “Mode 3. Selective photothermolysis.”

The Review can be accepted after serious editing.

Author Response

REVIEWER #3

The Review by David M. Harris and John G. Sulewski summarizes knowledge about phototherapy modalities. 5 modes of phototherapy are discussed including UVC, blue light, photothermolysis, vaporization and aPDT, in the light of the probability of inactivation of oral pathogens, especially Porphyromonas gingivalis. All these modes were characterized with light wavelength, mechanism of action, the depth of light penetration, selectivity and clinical trials.

The Authors provide a lot of useful information, like absorption spectra of bacterial cells, tissues and important substances as natural chromophores , dentin and so on. A lot of data on photoinactivation of oral pathogens and side effects and restrictions of these modes are also considered.

Response: Dear Colleague, Thank you for your time and the detailed review of our manuscript. It is always very useful to know what we did right.

Nevertheless, there are some shortcomings, and the first is about the presentation of the mechanism of UV killing.

Line 123.   Ultraviolet inactivation through mutagenesis.  

Line 127    mechanism of action: ultraviolet mutagenesis

Line 132   it is now accepted…    and references of 1980 and 1984 years.

Of course, UV is a powerful mutagen, and some mutations are lethal, but DNA damages like thymine dimers , DNA cross linking and others, are not mutations. If not repaired, they induce replication and transcription arrest, leading to the death of bacterion. To convert DNA damage to mutations repair processes are necessary, some of them being error prone, conducting trans- lesion synthesis . I recommend to refer5 to a fundamental book by E.C. Friedberg, G.C. Walker and W. Siede. DNA repair and mutagenesis. 1995. ASM Press, Washington, DC. Or more recent publications.

Response LINE 150-155: Thank you for this comment and for adding Friedberg to my library. Mutagenesis is not our long suit, but I hope our short exposure to Friedberg has improved the description.

Line 183   Mode 2: blue light inactivation through ROS generation (aBl)

       The title of the section is not a good fit. This section discusses not blue light, but rather the whole cell absorption spectra, and the effect of various wavelengths on bacteria employing their own chromophores as photosensitizers for ROS production.

Response LINES 45-55: Other reviews, both external and internal, have had difficulty with the labels of each mode and they have been modified to be more consistent.

Line 313     Two host tissues, the periodontal ligament and water…     What are two tissues?

Response LINES 343-344: The sentence is modified to read, “spectra for two of the host environments, the periodontal ligament (PL) and water.”

Line 317     The therapeutic ratio can estimated… The phrase is not correct, and 19 mg/ml = of what? All this block is not understandable, though the Authors describe their own experiments.

Response LINES 336-341: The introduction to this section is revised to improve clarity.

Line 437     Many photosensitizers tested on oral pathogenes are listed in this block without any references, and among them there are H2Owhich is not PS.

Response LINES 476-492: This section is revised to include 50 additional references. There were two studies that evaluated hydrogen peroxide as the PS, Mahdi et al (2015) and Odor et al (2020).

Line 447      advantage of aPDT is to avoid resistance. This is a controversial statement, because some groups have demonstrated the possibility of significant and inherited changes in susceptibility to aPDT and blue light after series of sublethal cycles. (See a review Rapacka-Zdo ´nczyk A, Wo ´zniak A, Michalska K, Piera ´nski M, Ogonowska P, Grinholc M and Nakonieczna J (2021) Factors Determining the Susceptibility of Bacteria to Antibacterial Photodynamic Inactivation. Front. Med. 8:642609.doi: 10.3389/fmed.2021.6426)

Response LINES 506-511: Rapacka-Zdonczyk et al present evidence of “tolerance” (changes in susceptibility) to aPDT and aBL following a series of sublethal cycles but did not consider it “resistance” since photoinactivation was still possible with an increased light dose. This statement is added to the text.

Line 470     They also tested for cytotoxicity of human gingival keratinocytes.  Inaccurate phrase.

Response LINE 532: The missing phrase improves the grammar.

Line 537 and beneath.     The lasers operating in Mode 4 are discussed in the section “Mode 3. Selective photothermolysis.”

Response LINE 607: Typo, should be: “operating in Mode 3”

Round 2

Reviewer 2 Report

I thank the authors for improving their manuscript.

Author Response

Thank you for your time and excellent suggestions.

Reviewer 3 Report

Dear Colleagues,

After the Author’s editing all my questions were answered and I am satisfied with all answers, except for one.

Line 486   The Authors insist that hydrogen peroxide is a photosensitizer (PS), and cite a questionable article by Mahdi et al, 2015, and more recent work by Odor et al, 2020 referring to the previous.     Hydrogen peroxide is not a photosensitizer, the molecule has not any chromophore and does not degrade under irradiation with blue light of enormous power (1144 mW/cm2 and 686 J/cm2) [Feuerstein O, Moreinos D, Steinberg D. Synergic antibacterial effect between visible light and hydrogen peroxide on Streptococcus mutans. J Antimicrob Chemother 2006;57(5): 872-876.] Mahdi et al cite this article incorrect and their conclusion about hydrogen peroxide is a PS may be disputed on the base of this article: H2O2 may strengthen the action of blue light, disrupting the cell membrane and also supporting intracellular natural photosensitizers with oxygene and ROS, thus enhancing killing of bacteria. Synergic effect between H2O2 and photosensitizers is well documented and the mechanism is depicted  in Sun, J.; Guo, Y.; Wang, Y.; Cao, D.; Tian, S.; Xiao, K.; Mao, R.; Zhao, X. H2O2 assisted photoelectrocatalytic degradation of diclofenac sodium at g-C3N4/BiVO4 photoanode under visible light irradiation. Chem. Eng. J. 2018, 332, 312–320. [Google Scholar] [CrossRef]

I believe, the Authors have made a large work and the Review has to be published, but they should better analyze this point.

Author Response

Thank you. Your argument is compelling. Hydrogen peroxide is no longer listed as a potential photosensitizer.